# The Astroglia Syncytial Theory of Consciousness

**DOI:** 10.3390/ijms26125785

**Published:** 2025-06-17

**Authors:** James M. Robertson

**Affiliations:** Independent Researcher, 508 27th Avenue South, North Myrtle Beach, SC 29582, USA; jimrobertsonmd@yahoo.com

**Keywords:** astrocytes and consciousness, astrocytes in memory formation, glia-neuronal interactions, theory of consciousness, astroglia syncytium and brain function, electrical and calcium signaling in the astroglia syncytium, panglia syncytium and information processing

## Abstract

The neurological basis of consciousness remains unknown despite innumerable theories proposed for over a century. The major obstacle is that empirical studies demonstrate that all sensory information is subdivided and parcellated as it is processed within the brain. A central region where such diverse information combines to form conscious expression has not been identified. A novel hypothesis was introduced over two decades ago that proposed astrocytes, with their ability to interconnect to form a global syncytium within the neocortex, are the locus of consciousness based on their ability to integrate synaptic signals. However, it was criticized because intercellular calcium waves, which are initiated by synaptic activity, are too slow to contribute to consciousness but ideal for memory formation. Although astrocytes are known to exhibit rapid electrical responses in active sensory pathways (e.g., vision), it was technically impossible to determine electrical activity within the astroglia syncytium because of the challenge of separating syncytial electrical responses from simultaneous neuronal electrical activity. Therefore, research on astroglia syncytial electrical activity lagged for over sixty years, until recently, when an ingenuous technique was developed to eliminate neuronal electrical interference. These technical advances have demonstrated that the astroglia syncytium, although massive and occupying the entire neocortex, is isoelectric with minimal impedance. Most importantly, the speed of electrical conductance within the syncytium is as rapid as that of neural networks. Therefore, the astroglia syncytium is theoretically capable of transmitting integrated local synaptic signaling globally throughout the entire neocortex to bind all functional areas of the brain in a timeframe required for consciousness.

## 1. Introduction

“It’s evident that the classical notion that information processing relies solely on neuron-to-neuron communication is essentially obsolete”, Agid and Magistretti 2021 [1].

The term tripartite synapse is currently used to acknowledge that astrocytes are active partners in synaptic functions in addition to traditional presynaptic and postsynaptic components [2]. Astrocyte peripheral process terminals extend into the synaptic cleft [3] or envelop individual excitatory and inhibitory synapses [4,5,6,7]. They also surround or are inserted into complex specialized groups of synapses [8].

Astrocytes modify excitatory and inhibitory synaptic signaling within each individual domain [9,10,11,12,13,14,15,16,17,18]. Subsequently, they process and integrate synaptic information [19,20,21,22,23,24,25,26,27,28]. They also regulate global neuronal networks through synaptogenesis and the elimination of synapses throughout one’s lifespan [29,30,31,32,33,34,35,36,37,38,39,40,41,42,43,44].

The most significant difference between astrocyte information processing and that of neurons is that astrocytes interconnect through innumerable gap junctions, electrically and chemically, to form a contiguous and continuous isoelectric syncytium throughout the neocortex. This massive structure assures astrocyte protoplasmic continuity throughout the brain and is known as the astroglia syncytium (AS) [45,46,47,48,49,50].

Gap junctions transmit integrated synaptic information throughout the AS, forming a complex network that continuously processes information originating from individual synapses. Thereafter, the AS is capable of the sensory, associative, and motor integration that is required for detecting, binding, and directing behavioral responses from and to the external environment through equivalent afferent and efferent synaptic signaling [51,52,53,54,55,56,57,58]. The representation of the external world, and subsequent behavioral responses, is the function of consciousness.

The major criticism of all neuron-based theories of consciousness is that it has repeatedly been demonstrated that perceptual sensory information is continuously subdivided and parcellated as it progresses within the neocortex [59]. Furthermore, the arrival of synaptic signals within subdivisions of brain functional areas does not coincide. This indicates a “disunity of consciousness” [60].

## 2. Astrocyte Domains, Microdomains, and Synaptic Interactions

Astrocytes express “virtually any type of receptor found in the CNS, which allows astroglia to perceive the neurochemical landscape of the nervous system”, Verkhratsky and Nedergaard 2018 [61].

It is widely accepted that rapid and complex bidirectional information occurs between synapses and perisynaptic astrocyte processes (PAPs) at each tripartite synapse [62,63,64,65]. PAPs constitute approximately 80% of the surface area of individual astrocyte domains, and their terminals are known as microdomains [66,67,68,69] (Figure 1).

Microdomains contain structural and neurochemical elements essential for information processing (glutamate and GABA neurotransmitters, neurotransmitter metabotropic and ionic receptors, and glutamate and ionic transporters), aerobic and glycolytic energy production (mitochondria and glycogen), motility (actin filaments), and intercellular communications (gap junctions) [1,7,25,28,70,71,72,73,74,75,76,77,78].

PAPs are compartmentalized. For example, there are two compartments related to different calcium responses evoked by synaptic activity, as well as the spatial relation of astrocyte NMDA receptors [25,74,75,76,77,78]. Additionally, messenger RNAs (mRNA) are confined to subcellular compartments where translation occurs [78].

### 2.1. Astrocyte-to-Neuron Signaling

Microdomains are extremely metabolically active [61]. They alter tripartite synapses by releasing a complex array of so-called gliotransmitters [79,80,81,82,83]. This includes the “classical” neurotransmitters glutamate and GABA, which are synaptically excitatory and inhibitory, respectively. The exocytic release of glutamate “controls synaptic strength” [84].

Importantly, D-serine is released by astrocytes, which is a necessary cofactor for NMDA activation at postsynaptic sites [85,86]. It is proposed that astrocytes control the release of D-serine, resulting in a temporal sliding scale that activates neuronal NMDA activation [87].

Microdomains also release other neuroactive molecules such as cytokines, purines, prostaglandins, and nucleotides, among others [80]. Astrocytes release glutathione, an amino acid, which ameliorates free radical damage to neurons [88].

There is also evidence that the perisynaptic net of extracellular matrix molecules is important for synaptic formation and functions. The perisynaptic net is secreted by astrocytic cell adhesion and matricellular molecules, as well as contributions from oligodendrocytes and microglia [89,90].

Specialized astrocytes in the mammalian retina, known as Müller cells, are essential for visual acuity. They function as “light guides” that transmit photons through the entire substance of the retina to activate one cone photoreceptor and, also in humans, ten rod photoreceptors. They act as “single-cell optical fibers” that “convey light directly to photoreceptors with minimal distortion or loss of photons” [91].

Lactate released from astrocytes promotes neuronal NMDA signaling [92] and is required for long-term memory formation [93]. Lactate is also increased during sleep, a time when memories are being consolidated [94]. Lactate is converted from glucose in astrocytes following the transport of glucose from cerebral capillaries via the AS [95].

Astrocyte-specific connexin 30 drives PAPs into synaptic clefts beginning on postnatal day ten, which continues throughout the lifespan of mice. It functions as an adhesive molecule and sets the basal synaptic scale [3]. Interestingly, 10–14 days is the time of visual (eye opening) and hearing awareness in postnatal mice.

### 2.2. Neuron-to-Astrocyte Signaling

A single axon passes through the territory of multiple astrocyte domains as it ramifies within widespread areas of cortical gray matter. Furthermore, each domain is influenced by multiple axons with diverse origins that express a variety of neurotransmitters and neuromodulators.

Synaptic glutamate is taken up by astrocyte glutamate transporters that convert it into glutamine. The glutamine is subsequently released to neurons for conversion back into glutamate, which is the essential transmitter for excitatory synapses. Neurotoxicity occurs unless glutamate and potassium are rapidly removed following synaptic discharge. This important function is performed by astrocyte microdomain glutamate and ionic transporters [96]. In addition to glutamate and GABA receptors, astrocytes express norepinephrine, serotonin, dopamine, acetylcholine, endocannabinoid, and opioid receptors [1,61,97]. Up to five receptors may be expressed in individual astrocytes [98].

Neuromodulators respond slowly compared to extremely fast-acting neurotransmitters. Among these, norepinephrine is known to ‘‘play an important role in defining what psychologist call the ego, the awareness we each have of being a distinct person separate from all others, confronting the universe on our own’’ [99]. Serotonin serves ‘‘an important role in engendering feeling states, such as love and hate, joy and sadness’’ [99]. Norepinephrine and serotonin receptors are known to coexist on individual astrocytes [100].

Stimulation of the locus coeruleus activates cortical astrocyte adrenergic receptors in vivo [101]. Astrocyte beta-adrenergic receptors induce cyclic AMP-generated pathways in response to catecholamines [102]. Astrocyte activation via endocannabinoids potentiates synaptic activity and is important in working memory [103].

An ultrastructural study confirms the presence of adrenergic synapses on astrocytes in addition to *en passant* terminals [104]. In fact, astrocyte adrenergic receptors are more abundant than those on neurons [102].

## 3. Astrocyte Domains Integrate Synaptic Activity

Microdomains integrate synaptic information from up to 2 million excitatory and inhibitory tripartite synapses within each human astrocyte domain [19,20,21,22,23,24,25,26,27,28]. Astrocytes incorporate synaptic information through the generation of elevated intracellular calcium transients and the induction of electrical currents.

### 3.1. Calcium Excitability

Neurons release glutamate and GABA to activate analogous receptors on astrocytes that increase intracellular calcium levels ([Ca^2+^]i), which occurs simultaneously with synaptic activation [105]. A recent in vivo study unequivocally demonstrates that whisker stimulation in live mice simultaneously increases astrocyte [Ca^2+^]i in barrel cortex astrocyte microdomains and astrocyte endfeet surrounding blood vessels [75].

There is no difference between the latency of astrocyte microdomains and neurons. Domain activity occurs 125 ms following synaptic activation. It was concluded that “astrocyte signal dynamics are sufficiently fast to influence cortical information processing and/or neurovascular coupling” [75].

Low synaptic activity induces small increases in [Ca^2+^]i, whereas stronger synaptic activity results in synchronous [Ca^2+^]i transients that, at higher levels of activity, form long-range calcium waves throughout the AS via gap junctions [1,19,45,50,61,106,107,108]. Intercellular calcium waves “generate signals that establish bridges between different neuronal networks, which then become functionally interconnected” … “because of their ability to conduct broad spatial interactions over long periods of time” [1].

Neural networks, however, are incapable of such long-term activation because of the very short lifespan of synaptic molecular components, which excludes them from persisting for the time required to encode and retain memories [51,109].

### 3.2. Electrical Activation

Electrical currents were first observed in astrocytes surrounding the optic nerve upon visual stimulation [110]. Subsequently, several groups have demonstrated that individual astrocytes respond with electrical membrane transients as rapidly as neurons [55,76,105,111,112,113]. Such rapid astrocyte responses are “compatible with a physiological role in fast activity-dependent synaptic modulation” [79].

Although decades of accumulating evidence demonstrate the importance of astrocyte calcium signaling, electrical signaling in astrocyte networks has lagged because of difficulties in eliminating interference from neuronal electrical currents. However, recent groundbreaking techniques have solved the problem of separating astrocyte electrical activity free of neural entanglement.

An ingenious technique demonstrates that electrical currents rapidly traverse a bridge of syncytial astrocytes from one isolated group of neurons to another. They note that such rapid electrical signaling globally could “bind” perceptive elements between associated areas of the neocortex. Such electrical activity occurs in a timeframe required for consciousness [114]. Additionally, enhanced electric activity may be “a prerequisite for memory formation or behavioral response” [55].

The entire AS is isoelectric with low impedance [115,116]. This allows the “entire syncytium to electrophysiologically behave as a singular unit” [117]. Interestingly, this is not limited to protoplasmic astrocytes but also extends to fibrous astrocytes that are interspersed in myelinated white matter tracts, including the corpus callosum, which interconnect the hemispheres. The cerebellar syncytium and the spinal cord syncytium, components of the AS, are also isoelectric [118]. Therefore, isoelectric connectivity with low impedance is a “general feature of astrocyte networks” [117].

Gap junctions also elicit electrical activity through the intercellular transport of ions [96]. Potassium is the most critical since it must be removed immediately after action potential discharges to avoid neurotoxicity and cell death, which occurs with prolonged retention within the synapse. Gap junction communication increases with higher potassium levels [119]. The rapid removal of potassium by astrocytes, and its discharge into the cortical vasculature, occurs simultaneously in all active tripartite synapses in the brain [50]. This is a major factor contributing to the isoelectric characteristics of the AS.

## 4. The Astroglia Syncytium Integrates Multiple Neural Networks Throughout the Brain

“Without the presence of the astrocytic syncytium and its capacity for integration, how else could the specificity of a signal be preserved amid this neuronal clutter”, Agid and Magistretti 2021 [1].

The AS is currently viewed as a complex multifunctional system of astrocytes connected by gap junctions composed of Connexin 43 (Cx43) or Connexin 30 (Cx30) hexamers (known as connexons or hemigap junctions). Connexons interlock with adjoining astrocytes to form a completed gap junction [45,46,47,48,49,50,120,121,122]. Gap junction heterogeneity correlates with underlying neuronal functional areas [123]. For instance, astroglia networks integrate within functional regions in the neocortex, such as the somatosensory cortex [124] and the glomerular layer of the olfactory bulb [125].

Gap junctions coalesce at the outer membranes of adjoining astrocytes to form tightly packed quasicrystalline clusters known as plaques (GJP). These are extremely dynamic and pleomorphic, assuming ever-changing shapes and numbers of gap junctions that can be visualized as they travel across the membrane surface.

Interestingly, astrocytes only form homotypic gap junctions (Cx43-Cx43 or Cx30-Cx30). Heterotypic (Cx43-Cx30) gap junctions do not form because of unique biophysical characteristics that inhibit heterotypic formation. Surprisingly, Cx43 and Cx30 GJs colocalize within the same GJP. However, they are segregated and do not perform overlapping functions. Indeed, 64% of adult rat brain astrocytes contain such mixed plaques [126].

Approximately 30,000 gap junctions connect rodent astrocytes [6]. However, there are vastly more gap junctions in human protoplasmic astrocytes, which are 2.5 times larger with 27 times the volume. PAPs are 10-fold more numerous with microdomains that are 2.6 times longer than those of rodents [66,67,68].

Each gap junction has a tightly sealed quasicrystalline intercellular channel that enables the transmission of molecules from 1 to 1.5 kDa in size. This includes molecules critical for information processing (e.g., glutamate, GABA, ATP, cyclic AMP, cytokines, messenger RNA (mRNA), short interfering RNA (siRNA), and microRNA (miRNA)), transmitting from one astrocyte to another, providing “a wealth of information to flow throughout the neocortex” [1]. “In principle, a small group of cells or possibly a single cell within a syncytium has the potential to affect gene expression in a larger group of cells or possibly an entire organ” [127].

Both siRNAs and miRNAs are particularly interesting because of their lifespan of several days and their influence on translational activities associated with innumerable targets, including those critical for cognition [78,127,128]. Both are gap-junction-specific. They only travel through gap junction networks composed of astrocytic Cx43.

Translation occurs primarily in microdomains [78]. siRNAs and miRNAs have “key roles in learning and memory” [129]. In fact, “siRNA can be delivered from the interior of one cell to that of another independent of the extracellular space and influence gene expression in the recipient cell” [127].

Syncytial astrocytes, by regulating synaptic connectivity, can “establish bridges between different neuronal networks, which then become functionally interconnected because of their ability to conduct broad spatial interactions over long periods of time” [1].

### Spatial Distribution of Astrocyte Domains

Uniquely, the AS exists in a three-dimensional and seamless confluence of regular polygon-shaped astrocyte domains [67,130]. The three-dimensional and seamless structure of the syncytium is perplexing. The functional significance of this peculiar structural organization is unknown.

Similarly, consciousness is also three-dimensional and seamless, which is evident *a priori.* This morphological equivalency has led to the proposal that this complex geometric matrix created by the global tiling of astrocyte domains provides a template for the expression of consciousness and explicit memories [24].

Over a century ago, William James noted a paradox related to consciousness. He stated that ‘‘there is no cell or group of cells in the brain of such anatomical or functional pre-eminence as to appear to be the keystone or center of gravity of the whole system” [131]. It is proposed that the AS is the “keystone” of “the whole system”. Furthermore, there is no empirical evidence that consciousness occurs in the laminae or minicolumns of cortical grey matter.

Mountcastle notes the major problem in extending minicolumns to general cognitive constructs: “How the patterns of neural activity involved in a sensory discrimination or categorization, distributed as they are in wide areas of the brain, are unified into perceptual wholes, and how they flow through to conscious experience, remain among the great enigmas in brain science” [132].

## 5. The Panglial Syncytium Integrates the Entirety of Brain Information

‘‘Glial cells are engaged in a global communication network that literally coordinates all types of information in the brain”, R. Douglas Fields 2014 [51].

Although Cx43 and Cx30 gap junctions are distributed in the cortex, coupling these gap junctions with other glia and vascular components expressing different gap junction connexins is important in the maintenance of the blood–brain barrier, neurovascular coupling, and action-potential-dependent myelin plasticity.

AS gap junctions with heterotypic connections to oligodendrocytes and vascular elements form the panglial syncytium (PS), which allows for the transfer of metabolic molecules to and from brain capillaries to maintain neuronal homeostasis. Less appreciated is their ability to control dynamic myelin plasticity based on action potential activity within white matter fiber tracts.

Astrocytes are the linchpins and integrators of this structure since they are the only cells that form universal gap junctional connections to all macroglia throughout the brain. This complex integrative system of multiple glial networks leads Fields to conclude that “glial cells are engaged in a global communication network that literally coordinates all types of information in the brain” and that “such oversight and regulation must be critical to brain function, and neurons are incapable of it” [51] (Figure 2).

### 5.1. Myelin Plasticity

Astrocytes Cx30 and Cx43 couple to oligodendrocytes-specific connexins to form heterotypic gap junctions (Cx43-Cx47 and Cx30-Cx32) [126,133]. The most abundant Cxs in these two cell types, Cx43 and Cx32, do not pair with each other [134,135].

Gap junction communication between astrocytes and axons in the nodes of Ranvier alter axonal signaling occurs even before the action potential arrives at the synapse by spreading electrical current [136].

Astrocytes sense ATP released by axons while action potentials travel along the axonal nodes of Ranvier [136]. Subsequently, they signal to oligodendrocytes by releasing leukemia inhibitory factor, which stimulates oligodendrocytes to increase myelin production [137,138,139].

Interestingly, the degree of myelination, determined by diffuse tensor studies, positively correlates with the intelligence of children and teens. Increased myelination is evident with learning [140,141]. For example, expert pianists exhibit increased myelination in areas related to this specialized ability [142].

### 5.2. The Corpus Callosum and Consciousness

“The appearance of the corpus callosum in the placental mammals is the greatest and most sudden modification exhibited by the brain in the whole series of vertebrated animals.” Thomas Huxley, 1863 [143].

The human corpus callosum is the largest myelinated white matter fiber tract in the human brain with over 200 million axons connecting the two hemispheres. Subsequently, the surgical ablation of the human corpus callosum resulted in the discovery of the bihemispheric specializations of specific brain functions.

For instance, the left hemisphere is verbal, analytical, arithmetical, and aware of details related to time and conceptual similarities. The right hemisphere is nonverbal and musical and aware of spatial relations and pattern recognition [144].

Sperry postulated that there are two forms of consciousness, one in each hemisphere. However, this is obscured in the right hemisphere because of its lack of verbal expression [145]. The existence of full consciousness is only possible with the extremely rapid bidirectional synergy between the two hemispheres via the corpus callosum. Therefore, the corpus callosum is viewed as a binding structure that rapidly incorporates the diverse mental functions of each hemisphere into consciousness [146].

On the molecular level, transcription factor SATB homeobox 2 (SATB2) specifies axon guidance via the corpus callosum, whereas transcription factor BCL11B/CTIP specifies axons to project within each hemisphere [147]. Both are directed to their appropriate placement by astrocyte attractant and repellent molecules through “glial slings” prior to the onset of perinatal synaptogenesis [148].

### 5.3. Neurovascular Coupling

Astrocytic endfeet directly contact the blood–brain barrier (BBB) through capillary endothelial cells and pericytes [1,95]. Astrocytic endfeet express the GLUT1 transporter, which facilitates the transfer of blood-borne glucose into the AS. Glucose is subsequently converted into nutrients, such as lactate and pyruvate, that are essential for neuronal survival [149] (Figure 3 and Figure 4).

Nutrients are distributed on an “as needed” basis, with the most active neurons receiving most essential metabolites (i.e., neurovascular coupling). Such “metabolic networks” sustain synaptic transmission [150]. Interestingly, these networks coincide with “metabolic” sodium waves within the AS [151,152].

Conversely, metabolic and ionic elements that are neurotoxic are transported through the AS and released into the general circulation. The most important electrolyte is potassium, which is neurotoxic if not immediately removed from the synapse following synaptic discharge. Since the greatest brain metabolic requirement is dedicated to maintaining synaptic activity, it is essential to rapidly remove potassium efficiently from the brain into the general circulation. In fact, removal is so efficient that it is a major contributor to the low-impedance and isoelectric characteristic of the AS [1,119].

### 5.4. Metabolic Pathways and Human Brain Evolution

RNA retrotransposons have incorporated novel genes into the human genome. The most important related to human metabolism is *Human-Specific Endogenous Retrovirus K (HTML variant) (HERV-K (HTML)* [153,154]. It was incorporated into the human genome approximately 1 million years ago following the separation of humans from Chimpanzees (approximately 9 million years ago).

*HERV-K (HTML)* resulted in a massive increase in mTOR (Mammalian Target of Rapamycin), a major metabolic pathway involved in protein synthesis and increased cell proliferation [153,154]. This was a major factor in increased human brain size and energy demands compared to other primates (e.g., Chimpanzees).

Increased mTOR activation in the human subventricular radial glia differentiates human embryonic brain corticogenesis from other primates. It is also important in human adult neurogensis [153,154]. Significantly, mTOR also regulates the translation of extensive novel mRNA targets with multiple intracellular functions in human brains [155,156].

## 6. Discussion

“Nonetheless, the presence of astrocytes is likely a necessary, if not sufficient, condition for explaining the existence of consciousness.” Agid and Magistretti 2021 [1].

The Syncytial Theory of Consciousness is a non-neuronal theory of consciousness. Stated simply, consciousness is expressed electrically throughout the global AS within hundreds of milliseconds, a time scale required for consciousness [114].

Memories, however, form over significantly longer timeframes. This is particularly true for long-term explicit memories, which must be encoded, retained, and reintroduced into consciousness over days, months, or years. Extremely complex calcium signals traveling through astrocytic networks are ideal for this function. The likely mechanism is the epigenetic modification of astrocyte DNA within astroglia networks. It is the extent of the astroglial syncytium, theoretically encompassing all cortical gray matter, that gives it massive computational power compared to current neurocentric models [65].

Galambos, over half a century ago, proposed that glia are essential components in human cognitive functions, including memory storage and innate and learned behavior. Initial ultrastructural studies of the brain, he notes, show a ‘‘huge collection of glia through which a nerve process occasionally wanders’’. Similarly, initial intracellular neuronal electrical activity does not support ‘‘behavioral patterns operating on a time base of hours, years, and generations” [157]. However, his hypothesis was ignored by neuroscientists.

Another ultrastructural study was the first indication that astrocytes are involved in cognition. It demonstrated increased synaptic contact by astrocytes in mice raised in an “enriched environment” compared to those raised in routine laboratory conditions [158].

Over the past three decades, gliobiologists have demonstrated that astrocytes are, either directly or indirectly, associated with cognitive functions long believed to be strictly neuronal [27]. Technical advances that distinguish glial activity from neuronal activity on a millisecond timeframe at the level of individual synapses, as well as the determination of the lifetime of synaptic molecules, have further eroded the concept of synaptic primacy in brain information processing.

The most conclusive empirical evidence to date is the transfer of human/primate-specific astrocyte exaptations into chimeric mice. Importantly, the transplanted human cells maintained their extremely large size and substantially increased calcium signaling, as observed in prior studies. Most remarkable was the marked increase in learning and memory exhibited in the chimeras compared to controls. Since consciousness is the basis of declarative memory and learning, it is reasonable to conclude that the chimeric mice were more conscious, with improved memory, than the normal control population [159].

To support a non-neuronal site for consciousness and memories, it is necessary to explain why the synapse alone, or distributed throughout the brain, cannot support these functions. The major obstacle undermining a neurological basis of consciousness is that perceptual research has increasingly demonstrated that the brain does not process sensory information in a unified manner but that such information is continuously subdivided or parcellated [59,60].

LeDoux states the overwhelming contemporary view of virtually all neuroscientists regarding the critical nature of synapses in cognition: “Most of what the brain does is accomplished by synaptic transmission between neurons, and by calling upon the information ended by past transmissions across synapses. Given the importance of synaptic transmission in brain function, it should practically be a truism to say that the self is synaptic. What else could it be?” [160]. Synaptologists are skeptical that the synapse alone would explain such a sweeping interpretation.

LeDoux’s conclusion ignores historical criticism regarding his synapse-only hypothesis. Sherrington, the discoverer of the synapse, was dubious that synapses were important in higher cognitive functions. He noted a paradox when extending his findings to the cognitive realm, as it would imply that each synapse can “both transmit and integrate information simultaneously”. He noted that “To pursue mind into the unicellular would be like looking for a firefly across astronomical distances” [161].

In evaluating the future of synaptology, it was noted that “It will also become increasingly apparent that synapses account for only a subset of the entire range of interactions that exist between neurons and glia and that contribute to their functional properties in mediating behaviour. Neuroglia contribute to the processing of information by neurons through a wide range of mechanisms traditionally viewed as nonneural and nonsynaptic” [162].

More refined methods to determine the lifetime of synaptic molecules have further eroded the concept of synaptic primacy. The postsynaptic density is replaced each hour [163]. Every forty seconds, actin molecules, required for dendritic morphological changes during synaptic plasticity, are produced, which have a half-life of ten minutes [164]. McCrone posits “how the heck do these synapses retain a stable identity when the chemistry of the cell is almost on the boil, with large molecules falling apart nearly as soon as they are made” [109]?

It is now evident that “the classical notion that information processing relies on neuron-to-neuron communication is essentially obsolete” [1]. Furthermore, it is virtually impossible to assign any form of memory, particularly long-term memory, as a function of the collective simultaneous actions of all synapses at any specific moment. Additionally, most synaptic functions are dedicated to a multitude of other functions, rather than mnemonic encoding.

The synapse can only be considered the penultimate stage of information processing in the brain. Sherrington was correct when he cited the paradox of the synapse, in that it could not possibly be both a signaling and integrating entity simultaneously. Prior to the Neuron Doctrine, the brain was considered to be a functional syncytium of neurons. Ironically, it is a syncytium, but of glia and not neurons.

## Figures and Tables

**Figure 1 ijms-26-05785-f001:**
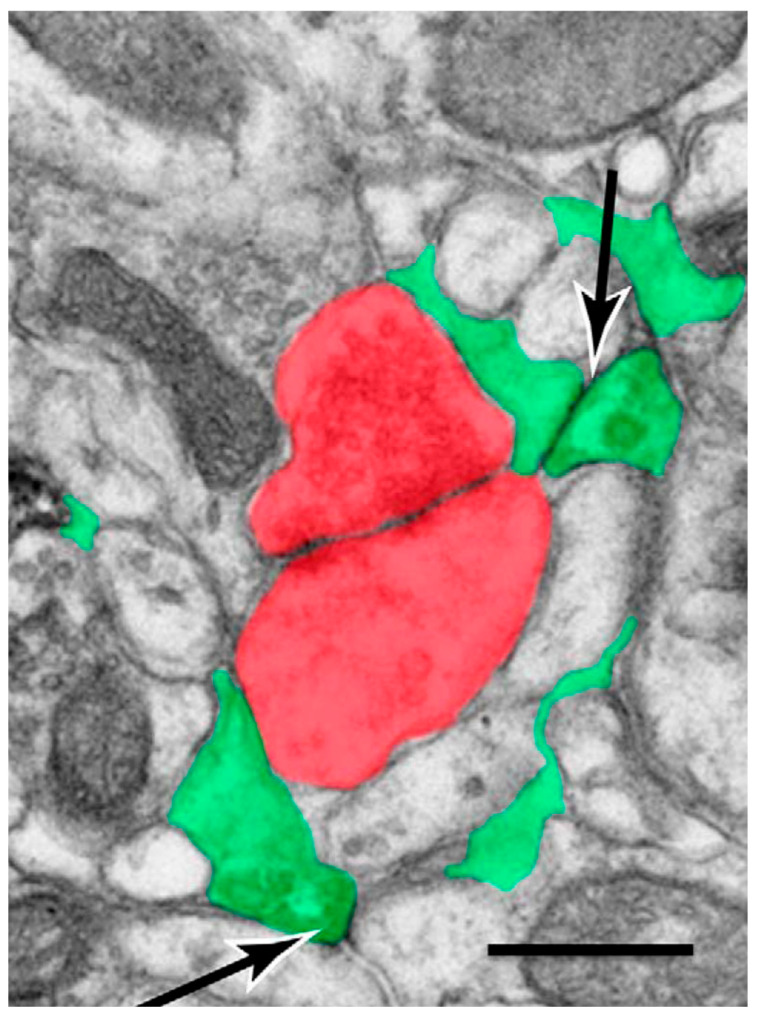
Astrocyte microdomain terminals (green) contact pre- and postsynaptic elements (red). Two gap junctions can also be observed connecting two adjacent astrocyte processes (arrows). (Modified from Giaume et al., 2022) [50].

**Figure 2 ijms-26-05785-f002:**
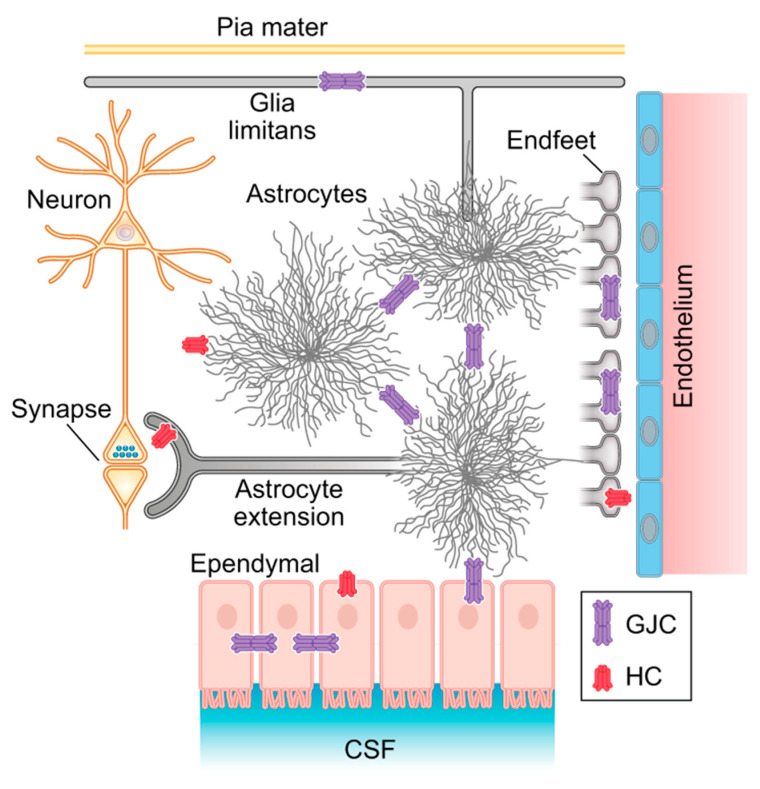
Astrocytes are the linchpin of the panglial syncytium: The homotypic gap junctions connect astrocytes to the astroglia syncytium, integrating excitatory and inhibitory synaptic information. Heterotypic astrocyte-to-oligodendroglia gap junctions function in myelin plasticity, whereas astrocyte endfeet are involved in neurovascular coupling. Additionally, astrocyte gap junctions coupled to ventricular ependyma monitor and respond to cerebrospinal fluid (CSF) signaling. (Modified from Giaume et al. 2022) [50].

**Figure 3 ijms-26-05785-f003:**
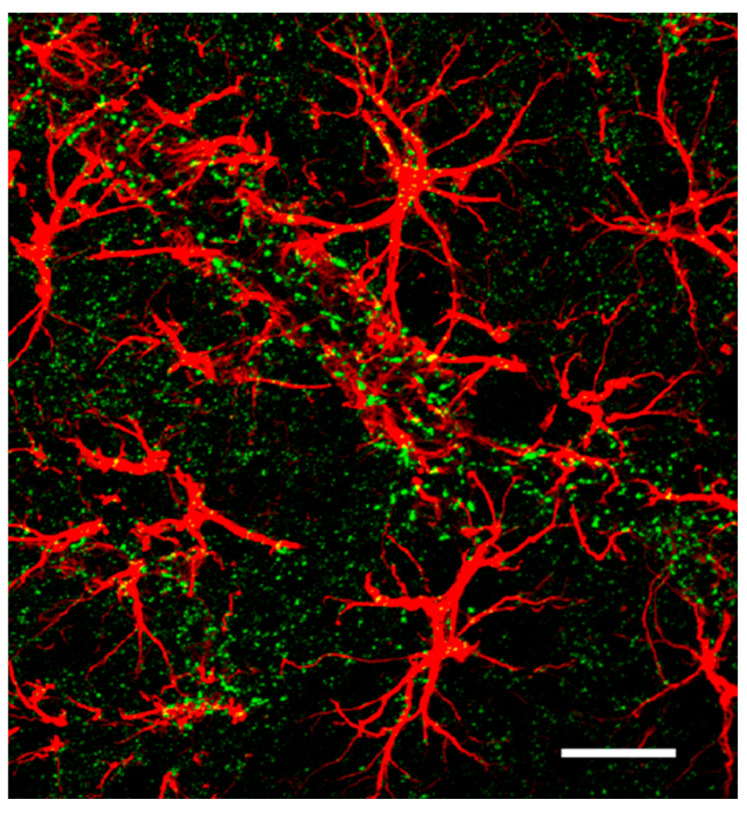
Astrocyte endfeet from multiple domains encircle cortical blood vessels (red). Cx30 (green) is expressed in the endfeet, as well as between and within astrocyte domains. (Modified from Giaume et al. 2022) [50].

**Figure 4 ijms-26-05785-f004:**
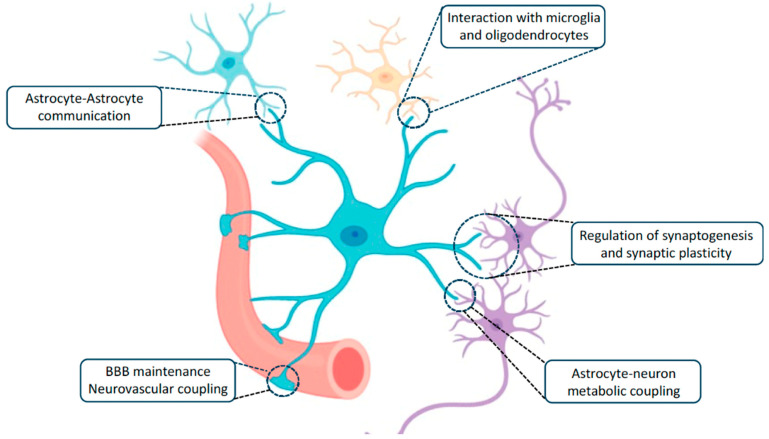
Astrocyte gap junction connections throughout the panglial syncytium forming the basis of myelin plasticity (astrocyte-to-oligodendrocyte heterotypic gap junctions). Astrocyte metabolic coupling and neurovascular coupling are also illustrated. Note that the astroglia syncytium is contained within the panglial syncytium. Individual astrocytes affect synaptic plasticity and conjoined with adjacent astrocytes to interconnect astroglia domains. (Modified from Escalada 2024) [56].

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
