# Peer review of "The Astroglia Syncytial Theory of Consciousness"

_ijms, 2025, doi:10.3390/ijms26125785_

Round 1
Reviewer 1 Report
Comments and Suggestions for Authors
This manuscript presents a comprehensive framework for exploring how interconnected glial networks, particularly astrocytes contribute to consciousness. It systematically examines the role of astrocytes in neural communication before delving into broader glial networks beyond astrocytes. Additionally, it explores specific aspects of glial function, such as myelin plasticity, neurovascular coupling, and metabolism.
I would like to highlight two points for discussion and potential revision:
1. Inclusion of a Diagram to Illustrate Glial Network Integration
Given the manuscript's extensive coverage of astroglial and panglial syncytium in relation to consciousness, incorporating a schematic diagram could enhance reader comprehension. Such an illustration would visually depict the structural and functional relationships among astrocyte domains, microdomains, and their integration into larger glial networks. This visual aid would help readers better grasp the complex interactions and the overarching narrative of the manuscript.
2. Clarification on Lactate Concentration During Sleep (Line 102)
The statement in line 102, "Lactate is also increased during sleep," appears to be inconsistent with several studies indicating that brain lactate levels decrease during sleep, particularly during non-rapid eye movement (NREM) sleep. For instance, research has shown that extracellular lactate concentrations decline during NREM sleep. Could the author give an explanation about it?
Author Response
Please note that I have submitted a new version of the manuscript that includes a new Abstract and corrected references.
A new article related to neurovascular coupling was released this week "Astrocytes at the heart of sleep: from genes to network dynamics" (2025) in Cellular and Molecular Life Sciences 82:207 https://doi.org/10.1007/s00018-025-05671-3. It comes from the Rouach group at College-de-France.
Reviewer 2 Report
Comments and Suggestions for Authors
The manuscript explores an intriguing and unconventional theory — the Syncytial Theory of Consciousness — positioning astrocytes, rather than neurons, as central to the emergence and maintenance of consciousness and memory. It brings together diverse literature to argue for a shift away from neurocentric paradigms toward a glia-centric framework, particularly emphasizing the human astroglial syncytium.
While the concept is novel and potentially valuable, the presentation is undermined by a number of structural, stylistic, and technical issues:
- The title lacks clarity and fails to reflect the astrocyte-centered focus of the paper.
- The abstract reads more like a series of bull et points or highlights, lacking cohesion and failing to engage the reader. Rewrite as a cohesive paragraph, beginning with a problem statement, outlining the novel proposal, and ending with the implications.
- The main text, though containing interesting content, is fragmented and dense. There is minimal use of visual aids, making it difficult to follow or remain engaged. Include at least one figure summarizing the central model or hypothesis. Consider adding a summary table contrasting neuron-based vs. astrocyte-based theories of consciousness.
- The discussion is relatively stronger, containing bold hypotheses and interesting perspectives, though at times it overstates the evidence or generalizes beyond the data. Many strong claims are made without adequate qualification or mechanistic depth. The discussion jumps between historical references, theoretical models, and experimental data without clear transitions. Moderate the tone of strong speculative statements unless clearly supported by evidence.
- Improve logical flow within paragraphs and between sections. Use topic sentences and transition phrases to guide the reader.
- The reference formatting is inconsistent, particularly with DOI styles (some start with “doi:”, others with https://).
Author Response
Thank you for your comments. I agree that the original manuscript was confusing, particularly the Abstract. Therefore, I have submitted a new manuscript with regular style and substance (whereas the original Abstract was a list of steps that laid the foundational empirical evidence of the theory).
Perhaps the title could be changed to "The Astroglial Syncytial Theory of Consciousness"
Regarding "visual aids", the article is theoretical and not amendable to visual aids. However, I have provided a Table of Contents that specifies the exact order and increasing complexity the reader will encounter. Hopefully, the publishers will incorporate it into the present article. To my knowledge, there is no visual illustrations in any of the 165 citations that would be sufficient for this purpose. This is meant to show the "cohesive pattern" that you refer to.
Regarding the references, I believe my original References were scanned and corrected by software at IJMS as the first list of references. It was beneficial in picking up one redundant Reference. This has been corrected in the new Manuscript submitted. However, their software incorrectly cited a variance that was not a true duplication, but actually different articles published in the same text (Glia 3rd Edition)
Regarding the discussion: As you know, all previous theories on consciousness are neurocentric. Therefore, a recent review of these is cited, since it is well beyond the scope of such a novel and iconoclastic theory to contrast with literally hundreds of neurocentric theories.
Additionally, since this is an extension of my original "Astrocentric Hypothesis (2002) and "Astrocytes Domains and the Seamless and Three Dimensional " explanation of conscious perception (2013), an historical perspective was chosen because the idea of consciousness as a glial process has not been declared explicitly in prior studies by gliobiologist. However, this is rapidly changing as empirical evidence is mounting related to astrocytes and Behavior.
Articles are often criticized because they only contain the most current references, lacking any less recent, but important empirical works. I believe the criticism by Sherrington regarding the paradox of the synapse being simultaneously a receptor and interpreter of information is important to show that, from the beginning, the synapse alone could not be important in higher cognitive functions. Additionally, I think it's important to validate Galambos assertion regarding the importance of glia in memory and behavior, as the hypothesis of such a prominent neuroscientist was completely ignored. However, the criticism of neurocentric-only interpretation by Shepherd, a preeminent Synaptologist, that forecast the importance of glia in synaptic functions was also significant and important.
Therefore, the discussion from Sherrington to Shepherd does constitute an advancement of - overtime - the very slow conversion of a synaptic neurocentric idea to one that allows an insight into contributions of other brain cells, particularly astrocytes.
The Neuron Doctrine is currently being questioned regarding it's tenets as well as ancillary ideas, such as LTP and LTD and the idea of Modules and Gamma Oscillators (please see my Gliocentric Brain (2018) article in this journal for a full accounting of these concepts that are entrenched even when such concepts have not been helpful regarding consciousness.
I hope these explanations are sufficient to correct the shortcomings you point out. Hopefully the new Abstract and Table of Contents will ameliorate your concerns. Thank you once again for taking the time to read the theory and I greatly appreciate your comments that significantly improves the article.
Round 2
Reviewer 2 Report
Comments and Suggestions for Authors
1. I didn’t see any tables in the new version, so I’m unable to comment on their content. I also couldn’t find where the tables were supposed to be inserted.
2. In any case, since this is an updated theoretical model, I think creating a simple schematic diagram shouldn’t be too difficult—perhaps by modifying the classic tripartite synapse illustration. You could highlight the involvement of astrocytes by adding a few labeled circles to indicate their roles in the process, and then use dashed lines to link them to functions like consciousness or others.
Author Response
I have sent the revised article outlined in yellow to the editor of the special edition (Ethan Xie). It includes a completly revised Abstract. It also originally had a Table of Contents that, for some reason, was deleted from my revised text.
The title was also changed to The Astroglia Syncytial Theory of Consciousness.
Once again, I don't think any visual aid would be helpful as the subject is not amendable to such. Additionally, I have always used Apple Text and currently using Microsoft for Mac to utilize doc references. If you insist on such, I will withdraw the article.
Reviewer 2:
I have managed to use Microsoft Word to add illustrations to the text. Therefore, I rescind my comments about withdrawing the article.
I have found 3 illustrations that are very good. I also just requested another to be formatted in the new edition. Furthermore, I have requested using another illustration of the "classical tripartite synapse". I am awaiting a response from the Oxford University Press to use it. However, although helpful, it is not essential.
The article now has a new Title, Abstract and uniform References in addition to the illustrations. I believe this fulfills all of your suggestions. Otherwise, the Body of the Manuscript is unchanged.
Thank you for your excellent suggestions. I am most grateful. These have resulted in a much improved "Astroglia Syncytial Theory of Consciousness"
Round 3
Reviewer 2 Report
Comments and Suggestions for Authors
There are no outstanding issues.